# Silicon Actuates Poplar Calli Tolerance after Longer Exposure to Antimony

**DOI:** 10.3390/plants12030689

**Published:** 2023-02-03

**Authors:** Eva Labancová, Zuzana Vivodová, Kristína Šípošová, Karin Kollárová

**Affiliations:** Institute of Chemistry, Slovak Academy of Sciences, Dúbravská cesta 9, 845 38 Bratislava, Slovakia

**Keywords:** antimony, antioxidant enzymes, nutrients, photosynthetic pigments, *Populus alba*, silicon

## Abstract

The presence of antimony (Sb) in high concentrations in the environment is recognized as an emerging problem worldwide. The toxicity of Sb in plant tissues is known; however, new methods of plant tolerance improvement must be addressed. Here, poplar callus (*Populus alba* L. var. pyramidallis) exposed to Sb(III) in 0.2 mM concentration and/or to silicon (Si) in 5 mM concentration was cultivated in vitro to determine the impact of Sb/Si interaction in the tissue. The Sb and Si uptake, growth, the activity of superoxide dismutase (SOD), catalase (CAT), guaiacol-peroxidase (G-POX), nutrient concentrations, and the concentrations of photosynthetic pigments were investigated. To elucidate the action of Si during the Sb-induced stress, the impact of short and long cultivations was determined. Silicon decreased the accumulation of Sb in the calli, regardless of the length of the cultivation (by approx. 34%). Antimony lowered the callus biomass (by approx. 37%) and decreased the concentrations of photosynthetic pigments (up to 78.5%) and nutrients in the tissue (up to 21.7%). Silicon supported the plant tolerance to Sb via the modification of antioxidant enzyme activity, which resulted in higher biomass production (increased by approx. 35%) and a higher uptake of nutrients from the media (increased by approx. 10%). Silicon aided the development of Sb-tolerance over the longer cultivation period. These results are key in understanding the action of Si-developed tolerance against metalloids.

## 1. Introduction

Antimony (Sb) is a non-essential metalloid whose concentration in the environment is rapidly increasing due to anthropogenic activity such as mining, smelting, waste incineration, and excessive use of commercial fertilizers [1,2]. This metalloid is also listed as a potential carcinogen [3] and several studies have already recorded its detrimental effects on human health [4,5,6]. Plants can take up Sb from the soil and accumulate it in their tissues, which can be linked with many negative impacts on plant growth and development. According to the studies of Herath et al. [7] and Shahid Muhammad et al. [8], the toxic concentration of Sb, which can disrupt the plant’s vital functions, is 5–10 mg Sb kg^−1^, but it depends on the plant variety or other important parameters such as the soil type, plant species, Sb speciation, redox potential, and availability [9,10].

In general, Sb in the environment is predominantly present in two forms—antimonate (SbV) or antimonite (SbIII)—while the second one is considered as the more reactive chemical form, easily absorbed and more harmful to all living organisms including plants [9,11,12]. The leakage of harmful amounts of Sb to the environment causes the inhibition of plant growth and biomass production as well as the inhibition of photosynthesis [13] and induces leaf chlorosis and necrosis [14]. Antimony might negatively influence the uptake of essential nutrients and disturb the synthesis of some metabolites [15]. It has also been reported that Sb can increase the peroxidation of membrane lipids [1] and cause severe oxidative stress [16,17]. Wang et al. [18] found that the highest Sb accumulation was in the roots of *Oryza sativa* L. The presence of Sb was also confirmed in the straws and grains of the plants. Accumulation of Sb in the plants can be a problem due to its entry into the food chain. It is well-documented that Sb has a negative effect on human health and causes diseases [6]. High amount of Sb in the human body increases immunological, neurological, reproductive, and developmental problems as well as the cancer of multiple organs [6,12]. Therefore, in order to minimize this threat, it is essential to find a solution of how to reduce the Sb concentration in plants and increase their defense mechanisms against this stress. One of the possible solutions might be the application of biostimulants, namely, plant growth hormones [19] or inorganic substances such as silicon [20].

Silicon is a quasi-essential element that might improve plant tolerance against toxic metals and metalloids through various mechanisms. Its positive impact on stressed plants is related to the stimulation of the antioxidant enzyme activity [21,22], enhanced root lignification [10], and increased endodermal cell wall thickness, which can block the entry of the heavy metal into the cells [23]. Furthermore, it has been suggested that Si in the optimal concentration decreases uptake, translocation, and increases the chelation of heavy metals by plants; subsequently, this improves the plant growth and tolerance against stress [24]. Multiple studies have confirmed that Si increases the fitness of plants [25,26,27]. One of the ways in which to gain new knowledge about Si action during the improvement of plant tolerance to Sb is to examine the influence of Si and Sb in plants tolerant to heavy metals.

The trees of the Salicaceae family are known for their ability to grow in a contaminated environment and for their increased heavy metal accumulation in the tissues. Poplar trees and poplar callus have been used for stress studies multiple times [28,29] because of their unique response to stressors. Calli cells are not lignified; hence, they provide a better option to study the cell reaction to the application of Sb and Si. Therefore, the poplar calli prepared by the in vitro technique are convenient for the study of the changes at the tissue level and the evaluation of the defensive mechanisms of plants grown under stress conditions.

In this study, the poplar callus (prepared from *Populus alba* L. var. pyramidallis) was used to determine the involvement of Si in the defense mechanisms against Sb stress. The main aim of this study was to clarify the influence of Si on callus tissue stressed by Sb over time (short and long cultivation). To elucidate the Sb and Si effects, the concentration of Si and Sb (1); the fresh and dry mass (2); the concentration of the photosynthetic pigments (3); the activity of the antioxidant enzymes (4) as well as the concentration of the selected mineral nutrients (5) in the poplar callus were determined.

## 2. Results

### 2.1. Effect of Sb and/or Si Treatments on the Concentration of Si and Sb

The poplar calli were cultivated in three subcultures and exposed to Sb and/or Si to obtain the results of the short (one subculture) and the long cultivation (three subcultures). Every subculture lasted for three weeks (Figure 1).

The poplar callus cultivated on the media containing only Si (the Si treatment) contained higher doses of Si than the calli cultivated on the media containing Si and Sb (the Sb + Si treatment) (Figure 2). Furthermore, the Si treatment accumulated more Si in the long cultivation than in the short cultivation (the concentration increased by 17.9%). However, the concentration of the Si in the calli of the Sb + Si treatment decreased by 20.6% after the short cultivation, and by 34.8% after the long cultivation, compared to the Si treatment. No significant difference in the Si concentration between the short and the long cultivation of the Sb + Si treatment was recorded.

Poplar callus of the Sb treatment accumulated a higher amount of Sb (by 9.6%) after long cultivation than after short cultivation (Figure 3). However, the cultivation duration had no effect on the Sb concentration in the Sb + Si treatment. The concentration of Sb in the Sb + Si treatment decreased after the short cultivation by 32.2% and after the long cultivation by 35.8% compared to the Sb treatment.

### 2.2. The Growth of the Callus Exposed to Sb or/and Si

After both the short and long cultivation, the Sb treatment negatively affected the fresh and dry mass of the poplar callus (Figure 4 and Figure 5). After the short cultivation, Sb inhibited the fresh mass by 38.4% and the dry mass by 35.9% compared to the control. Similar results were obtained after the long cultivation—the fresh mass decreased by 30.6% and the dry mass by 34.8%. However, the application of Si to media alleviated the negative effect of Sb. We determined that the fresh mass and dry mass of the Sb + Si treatment increased after the short cultivation by 35.8% and 31.4%, respectively, and after the long cultivation by 33.4% and 46.3%, respectively. Even though the subculture for both the short and long cultivation lasted for three weeks (21 days), we determined a significant difference between the dry mass of the Sb + Si treatment after the short and long cultivation. The longer exposure to Sb and Si (the long cultivation) increased the dry mass of the calli compared to the short exposure (the short cultivation).

### 2.3. The Influence of Sb and/or Si on the Concentration of Photosynthetic Pigments

The application of Sb to the callus media decreased the concentrations of photosynthetic pigments (chlorophyll *a*, chlorophyll *b*, and carotenoids) both in the short and long cultivation compared to the calli cultivated on basal media (Table 1). The concentrations of chlorophyll *a*, chlorophyll *b*, and carotenoids detected in the calli of the Sb treatment were lower than in the control by 67.0%, 68.5%, and 64.2%, respectively, after the short cultivation, and by 78.5%, 74.5%, and 59.1%, respectively, after the long cultivation. Similar to the improved callus growth, the Si application on the Sb-stressed callus (the Sb + Si treatment) improved the concentrations of chlorophyll *a* by 163.5% and chlorophyll *b* by 162.2% after the short cultivation compared to the Sb treatment. However, the positive effect of this treatment was more evident after the long cultivation because the concentrations of chlorophyll *a* and chlorophyll *b* increased by 275.3% and 267.5%, respectively. In addition, the concentrations of the carotenoids were raised in the Sb + Si treatment by 121.7% after the short and by 116.5% after the long cultivation. The length of the Sb and/or Si exposure influenced the contents of the pigments. The significant increase in chlorophyll *b* concentration between the short and long cultivation was determined in the control, the Si, and the Sb + Si treatment. Only the Si treatment contained higher concentrations of chlorophyll *a* after the long than after the short cultivation. In addition, the concentration of carotenoids was decreased in the control, the Si, and the Sb + Si treatment in the long cultivation compared to the short cultivation.

### 2.4. Effect of Sb and/or Si on the Activity of Antioxidant Enzymes

The presence of Sb in the medium also affected the activity of antioxidant enzymes in the poplar callus (Figure 6). The activity of all observed antioxidant enzymes (SOD, CAT, and G-POX) was significantly increased in the Sb treatment compared to the control. After the short cultivation, the activity of SOD increased by 24.8%, CAT by 91.5%, and G-POX by 24.0%, and after the long cultivation, the activity increased by 21.2%, 74.0%, and 42.2%, respectively. However, the addition of Si to the media containing Sb (the Sb + Si treatment) significantly decreased the activity of enzymes both after the short (SOD by 16.8%; CAT by 20.0%; G-POX by 19.5%) and after the long (SOD by 14.8%; CAT by 17.7%; G-POX by 30.1%) cultivations compared to the Sb treatment. The length of the cultivation did not have a significant effect on the activity of the enzymes.

### 2.5. Effect of Sb and/or Si on the Concentration of the Mineral Nutrients

The mineral nutrient concentrations were also negatively influenced by the addition of Sb to the callus media (Table 2 and Table 3). The Sb treatment significantly decreased the concentration of macronutrients after the short cultivation (P by 16.1%, K by 10.6%, Ca by 8.2%, and Mg by 9.7%) and after the long cultivation (P by 21.7%, K by 11.2%, Ca by 9.9%, and Mg by 13.0%) compared to the control. The Si treatment affected only the concentration of Mg after the long cultivation (increased by 5.4% compared to the control). Similar to the other parameters determined, the macronutrient concentrations were influenced by the Sb + Si treatment. The macronutrient concentrations in the calli of the Sb + Si treatment increased after the short (P by 14.9%, K by 7.3%, Ca by 6.9%, and Mg by 9.3%) and after the long cultivation (P by 13.4%, K by 9.3%, Ca by 8.6%, and Mg by 10.6%) compared to the Sb treatment. Moreover, the longer exposure of calli to Si (the long cultivation) increased the uptake of Ca compared to the shorter exposure (the short cultivation). 

The changes caused by the different callus treatments in the micronutrient concentrations displayed similar trends as the macronutrient concentrations. The Sb treatment lowered the concentrations of micronutrients after the short cultivation as follows: Fe by 16.9%, Mn by 16.6%, Zn by 20.2%, and Cu by 15.4% compared to the control. Likewise, after the long cultivation, the Sb treatment lowered the micronutrient concentrations as follows: Fe by 13.6%, Mn by 21.8%, Zn by 22.5%, and Cu by 26.7% compared to the control. The positive effect of Si on the Sb-stressed calli (the Sb + Si treatment) was also evident in the micronutrient concentrations. The Sb + Si treatment increased the concentrations of Fe by 13.9%, Mn by 14.5%, Zn by 21.9%, and Cu by 18.2% after the short cultivation compared to the Sb treatment. After the long cultivation, the Sb + Si treatment raised the concentrations of Fe by 30.5%, Mn by 7.3%, Zn by 18.5%, and Cu by 27.3% compared to the Sb treatment. The addition of Si to the media improved the uptake of micronutrients after the long cultivation compared to the short one. The micronutrients most affected were Mn and Cu. However, the Si treatment did not significantly affect the concentration of the micronutrients compared to the control. 

## 3. Discussion

Toxic metalloids negatively affect plant growth and development [30]. However, some metalloids such as the micronutrient—boron—might improve the development of callus; therefore, its addition to the media has been suggested for the propagation of plants in vitro [31]. Even though boron is important for cell wall synthesis and membrane stability, etc. [32], its high concentration in the callus media causes necrosis of the callus tissue and decreases the proliferation rate [31]. Yang et al. [33] studied the responses of a tolerant *Pteris vittata* and susceptible *Arabidopsis thaliana* calli to the metalloid—arsenic—in the media. They found that susceptible *A. thaliana* callus turned brown and the viability of cells decreased.

Here, we studied and discussed the effects of the metalloid antimony in poplar callus tissue. Two traits that make antimony especially dangerous for plants are its high reactivity and its two oxidation states: Sb(III) and Sb(V). In the cells, Sb displaces other metals from their binding sites or blocks functional groups, thereby altering the biosynthesis of numerous compounds. In the present study, we confirmed that Sb had negative effects on the poplar callus such as the decrease in fresh and dry mass, the decrease in the photosynthetic pigment concentrations as well as the reduction in the mineral nutrient concentrations. However, Sb increased the activity of antioxidant enzymes, which confirmed that Sb induces oxidative stress. The biostimulants of various chemical and biological compositions (including organic and inorganic compounds) were tested to elucidate their potential to alleviate the impact of heavy metal stress on plants [30,34]. The tested organic biostimulants are different types of phytohormones such as auxins, cytokinins, abscisic acid, and gibberellins [30], while Si, Se, Co, and Al are commonly investigated inorganic stimulants [34]. It is very important to mention that the positive effects of all biostimulants depend on the concentration used. In this study, we focused on the Si, which is a quickly emerging biostimulant for plants, mainly for its usage in plant defense against a wide range of stress factors [35,36].

First, to elucidate the impact of Sb and Si on the poplar tissue, the Sb and Si uptake by the tissue had to be determined. The concentration of Sb and Si in the tissues can provide information about the extent of Sb absorption and accumulation by the calli tissue. Poplar plants are known as good accumulators of toxic metals including heavy metals and the poplar callus tissue has gained similar traits [28,37,38]. In the present study, the Sb treated calli contained a higher concentration of Sb after the longer cultivation period, while maintaining stable biomass production. This suggests that poplar calli are good accumulators of Sb and have some degree of Sb-tolerance. The application of Si to the media decreased the uptake of Sb by the poplar callus tissue and the concentration of the accumulated Sb was almost the same after the long cultivation than after the short cultivation.

The effects of Sb on the fresh and dry mass of poplar callus were similar to the effects of Cd or As on poplar callus tissue, which were determined in our previous studies [28,29]. In our previous studies, we used the same cultivation conditions and experimental design as in this study; however, the calli media were supplemented with a significantly higher concentration of Sb than Cd or As. Poplar trees are well-known for their ability to tolerate heavy metals [37,38]; however, less information is available on their ability to tolerate metalloids. Although poplar callus was not able to develop the tolerance to Sb independently after the longer cultivation as opposed to the experiment with Cd [28], the callus retained its vitality on a similar level after the longer exposure to Sb, likewise as in the case of As [29]. Furthermore, the callus tissue was able to actuate its antioxidant system, which suggests a good level of Sb tolerance.

Silicon positively affects the growth parameters of the plants as well as calli cultivated in substrates polluted by many different toxic metals [1,28,29]. The addition of silicon to the substrate increases the root and shoot growth of the plants, improves the biomass production, and promotes the photosynthesis via the increase in the photosynthetic pigment concentrations. The direct action of Si during stress is not known; however, its positive effect on plant growth and metabolism is probably indirect via improving the detoxification mechanisms [39]. As above-mentioned, the biostimulant properties of Si depend on the concentration used. Hence, our previous study [28] focused on the identification of the most efficient Si concentration for the alleviation of the negative effects of Cd^2+^ on the poplar callus. We found that both the 2.5 mM and 5 mM concentrations effectively improved poplar tolerance to Cd^2+^ during the short cultivation. However, only the 5 mM concentration significantly improved the tolerance to Cd^2+^ during the longer cultivation. The type of Si, environmental stress, and plant species also affects the effects of Si application. For example, the growth stage of the plant during which it comes into contact with both the heavy metal and the biostimulant has an impact on the effectivity of the growth stimulation. Nwugo and Huerta [40] reported that *Oryza sativa* plants stressed by Cd reacted more positively to Si supplementation in the later stage of growth than in the early stage. In the present study, the addition of Si to the media significantly improved the dry mass of the Sb-stressed callus over the longer cultivation, which suggests silicon’s supporting role in the development of Sb-tolerance.

Low concentrations of ROS are necessary for plants because they act as signal molecules that regulate plant growth and development [41] However, their overproduction is linked with abiotic and biotic stresses and can be detrimental for plant tissues [42]. In the present study, Sb induced oxidative stress and increased the activity of all of the selected antioxidant enzymes. The application of Si in the media of Sb-stressed callus improved the poplar callus vitality and influenced the uptake and sequestration of Sb within the cells. One of the detoxification strategies utilized by the Si is the activation of the antioxidant defense system, which reduces the oxidative damage caused by the high concentration of ROS [39]. A lower concentration of Sb in the callus tissue probably lowered the activity of the antioxidant system, which also suggests lower oxidative stress. A positive effect of Si on the plants and calli treated with different heavy metals or salinity was also determined by Vaculíková et al. [1], Labancová et al. [28], and Yan et al. [43].

One of the general plant defense mechanisms against higher concentrations of toxic metals in the environment is the modification of the cell wall composition by processes such as suberization and lignification [44,45]. This process, regulated by enzymes such as peroxidases, prevents the uncontrollable uptake of the toxic metal, and in turn, protects the membranes against damage [46]. However, the callus tissue prepared from leaves or fruits is composed of cells with only the primary cell wall, hence, the callus cells do not have another protective barrier against toxic metal stress [28,47]. The damaged membrane components change the membrane charges and might prevent the uptake of nutrients [48,49]. In the present study, Sb significantly decreased the accumulation of nutrients, which might be connected to the induced oxidative stress. However, in the present study, Si promoted the uptake and accumulation of essential macronutrients and micronutrients, except for Ca in the short cultivation. The Si induced amelioration of negative effects caused by heavy metals can be connected to the enhanced uptake and accumulation of essential nutrients [39]. It is known that during stress caused by nutrient deficiency or toxic metal presence, Si alters the expression of transporter genes responsible for the uptake of different nutrients [27]. For example, Si upregulated the expression of P transporter genes (*9TaPHT1.1* and *TaPHT1.2*) in the *Triticum aestivum* L. plants cultivated in nutrient-deficient soil [50]. In addition, Schaller et al. [51] found that Si improved the mobility of P in the soil, increasing its solubility, and thus the uptake. Furthermore, the expression pattern of Fe-related genes (*NAS1*, *YSL1*) in the *Cucumis sativus* plants cultivated in limited iron supply conditions was changed by the addition of Si to the plants [52]. The supplementation of callus media with an appropriate concentration of Si improved the membrane stability of the Cd-stressed callus cells [49] and increased the nutrient uptake. Similar results of nutrient uptake were obtained in the As-stressed calli [29]. In this study, the lowered activity of antioxidant enzymes in the Sb + Si treatment suggests a reduction in the oxidative stress and the prevention of ROS-induced membrane damage, resulting in the improved nutrient uptake. In addition, the lower amount of Sb taken up by callus tissue also prevented the excessive deterioration of the nutrient transport systems.

The improved biomass production by the Sb + Si treatment can be connected to the increased uptake of nutrients that take part in various growth and development processes. The calli of the Sb + Si treatment accumulated the macronutrients—P and Mg—in much higher concentrations than the calli of the Sb treatment (increased from 9.3% up to 14.9%). These macronutrients are part of numerous enzymes and are involved in defense mechanism processes [53,54]. Phosphorus influences nitric oxide generation, which is linked to proline metabolism, the ascorbate-glutathione system, and glyoxalase system [55]. Magnesium increases the synthesis and exudation of organic acid anions or regulates the proton-ATPase activity and cytoplasmic pH [54].

Silicon supplementation more significantly alleviated the negative effects of Sb on the uptake of micronutrients. Some micronutrients are part of the antioxidant enzyme molecules and other metalloenzymes, but also act as catalysts of many reactions and take part in the osmoregulation processes and plant defense against stress [56,57]. Zinc maintains membrane integrity, the structural integrity of proteins, and the synthesis of plant hormones. Iron reduces electrolyte leakage, membrane damage, and forms an Fe–S cluster, while Cu protects the cellular membrane and reduces lipid peroxidation. 

Two mechanisms of the action of Si have been described in plants thus far [39]: (i) the deposition of Si in the cell walls, blocking the entry of heavy metals to the cells while directing the uptake to the apoplasmic pathway [58]; (ii) the formation of the heavy metal–Si complexes [39,59], which can form in the medium and in the plants. A similar impact of Si supplementation on the uptake and accumulation of different heavy metals as shown in this study was found in *O. sativa* [60]. In the case of Sb and poplar callus tissue, the Si application decreased the accumulation of Sb, and improved the plant biomass by increasing nutrient uptake and the concentration of photosynthetic pigments. Hence, both of the above-mentioned mechanisms of Si–Sb (similarly to heavy metals) interaction were probably implemented in the callus tissue. The decrease in antioxidant activity in relation to Si application and over longer exposure also suggests lowered oxidative stress in the tissue and the development of higher tolerance to Sb. These results aid the development of biostimulants that could not only improve the biomass production of stress susceptible, but also tolerant species used in phytoremediations.

## 4. Materials and Methods

### 4.1. Plant Material Cultivation

In our experiments, we used poplar callus prepared from the one-year-old shoots of white poplar (*Populus alba* L. var. pyramidallis) cultivated in three subcultures, similarly to the methods of Labancová et al. [28] and Kučerová et al. [29], with the addition of antimony (in the concentration 0.2 mM and the form potassium antimony(III) oxide tartrate) and silicon (in the concentration 5 mM and the form of sodium silicate solution) into the basal medium. The basal medium was agar medium [61] supplemented with NAA (2.5 μM), IAA (5.5 μM), 2,4-D (0.5 μM), and sucrose (30 g l^−1^). The material for the analyses was sampled after 3 and 9 weeks of cultivation (the short and the long cultivation) in four treatments: control, silicon (Si), antimony (Sb), and a combination of antimony + silicon (Sb + Si). The procedure is described in detail in Figure 1.

### 4.2. Determination of Sb and/or Si Concentrations 

The concentration of Sb and/or Si was determined from the dry material (72 h, 45 °C) collected after the short and long cultivation (see Figure 1). The samples were prepared by the method described in Šípošová et al. [62] and the analyses were carried out by flame atomic spectrometry (AAS Perkin Elmer 1100 and 4100) and by inductively-coupled plasma mass spectrometry (ICP-MS, Thermo iCap Q) at the Institute of Laboratory Research on Geomaterials, Faculty of Natural Sciences, Comenius University in Bratislava, Slovakia.

### 4.3. Determination of the Fresh and Dry Weight

Erlenmayer flasks with the medium (m_e_) were weighed before the inoculation and weighed afterward with the inoculum (m_e+i_); the following formula was used—m_i_ = m_e+i_-m_e_—and the mass was adjusted to 1–1.5 g. To obtain the fresh mass of the callus after 3 and 9 weeks of cultivation (the short and long cultivation), the callus mass was carefully removed from the media and weighted. The removed callus mass was left in the desiccator (72 h, 45 °C) to obtain the exact dry weight.

### 4.4. Determination of the Photosynthetic Pigment Concentration

The concentration of photosynthetic pigments was determined from fresh material collected after the short and the long cultivation, extracted with 80% (*v*/*v*) acetone, and homogenized in a cold mortar with a mixture of sea sand and MgCO_3_. The samples were centrifuged (11 min, 8460 g, 4°C) and the concentrations of chlorophyll *a*, chlorophyll *b*, and carotenoids were measured spectrophotometrically from the supernatant and calculated based on the formulas of Lichtenthaler [63].

### 4.5. Determination of the Antioxidant Enzyme Activity

The material for the determination of the enzyme activity was collected after the short and the long cultivation and the samples were prepared according to the method of Šípošová et al. [62]. The concentration of the soluble proteins was determined by the Bradford method using the bovine serum albumin as a standard [64]. In our experiments, the activity of three antioxidant enzymes in poplar callus was determined: superoxide dismutase (SOD, E.C.1.15.1.1), catalase (CAT, E.C. 1.11.1.6), and guaiacol peroxidase (G-POX, EC 1.11.1.7), according to the methods of Madamanchi et al. [65], Hodges et al. [66], and Frič and Fuchs [67], respectively.

### 4.6. Determination of the Mineral Nutrient Concentrations

The concentration of the selected mineral nutrients (Ca, Fe, K, Mg, P, Zn, Cu) was determined to be similar to that described in Section 4.2 “Determination of Sb and/or Si concentrations”. 

### 4.7. Statistical Analyses

The data are displayed as a mean value ± standard error (SE). The experiments were repeated at least three times and the differences between the experimental groups were evaluated by an analysis of variance (ANOVA) and the Tukey test at *p* ˂ 0.05 with Statistica, Version 9.1 (StatSoft, Tulsa, OK, USA).

## 5. Conclusions

The heavy metal tolerance by the trees of the Salicaceae family has been in the spotlight over the last decade; however, their response to metalloids such as Sb on a more detailed level has not been established yet. In the present study, the poplar callus tissue did not independently develop tolerance against the increased Sb(III) concentration in the media. The concentration of Sb in the callus tissue was high and its presence in the cells actuated the antioxidant enzymes. The higher activity of SOD, CAT, and G-POX found in our study suggests the induction of oxidative stress in the callus. Furthermore, the nutrient uptake was significantly disturbed, and the concentration of photosynthetic pigments decreased, which can all be linked to low biomass production. However, the application of the Si in the 5 mM concentration to the media of Sb-stressed callus improved the biomass production, and this positive effect was more evident after the longer cultivation. Silicon application considerably decreased the uptake of Sb to the callus and improved the oxidative status of the tissue. The lower activity of antioxidant enzymes can be linked to lower levels of ROS and an increased uptake of nutrients to the tissue. The higher doses of nutrients such as P, K, Mg, Ca, Fe, Cu, Zn, and Mn not only aided the biomass production directly, but also improved the activity of the antioxidant enzymes. Moreover, the increased photosynthetic pigment concentrations in callus tissue also supported the biomass production. Hence, the addition of Si to the media promoted the development of Sb-tolerance in poplar callus. These results suggest that the application of silicon on Sb tolerant plants used for the remediation of contaminated areas could be an effective method to enhance their efficiency.

## Figures and Tables

**Figure 1 plants-12-00689-f001:**
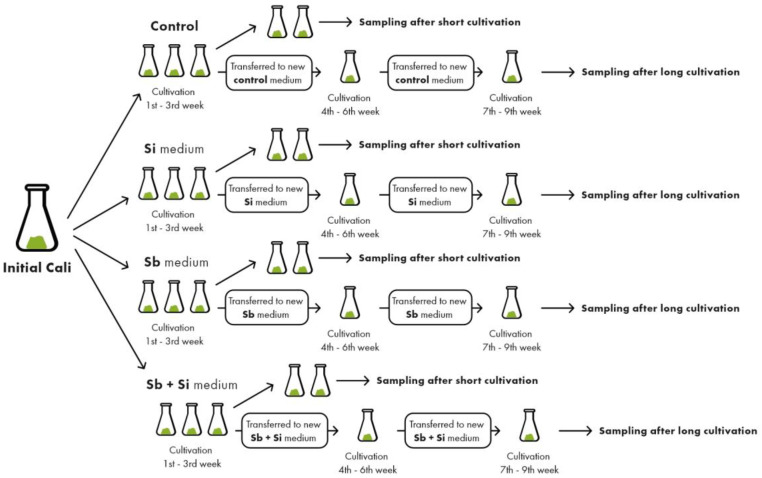
The scheme of experimental design. The initial callus material was first inoculated into the control, Si, Sb, Sb + Si media (15 callus samples for every treatment). The weight of the inoculum at the beginning of each cultivation was 1–1.5 g. For the effects of short cultivation, 10 calli from every treatment were sampled after 3 weeks of the cultivation. The rest of the callus samples (five for every treatment) was transferred into a new media (control treatment to control media, Si treatment to Si media, Sb treatment to Sb media, and Sb + Si treatments to Sb + Si media). At the end of the sixth week of the cultivation, the calli were again transferred into a new control, Si, Sb, Sb + Si media (10 callus samples for every treatment). For the effects of a long cultivation, 10 calli from every treatment were sampled after 9 weeks of the cultivation. Each growth experiment, repeated three times on separate days, consisted of 30 callus samples per treatment.

**Figure 2 plants-12-00689-f002:**
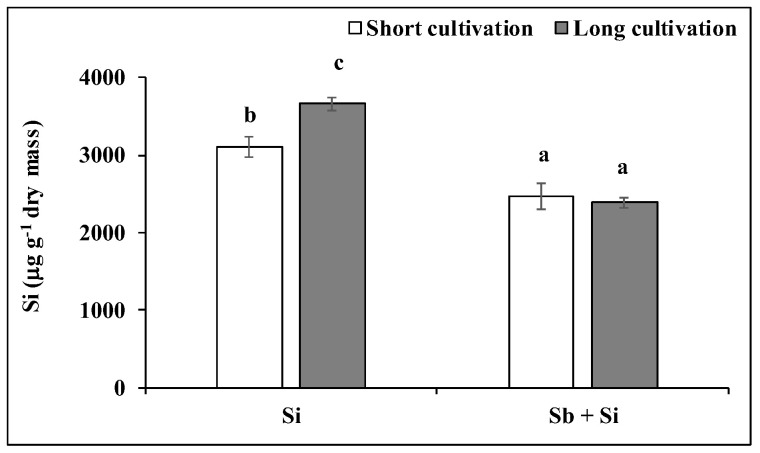
Concentration of Si (μg g^−1^ dry mass) in poplar calli after the short and long cultivations. The data are presented as the means ± standard error (*n* = 3). Si treatment—basal medium + Si (5 mM); Sb + Si treatment—basal medium + Sb (0.2 mM) and Si (5 mM). Different letters above bars denote statistically significant differences in the parameters between the treatments at *p* < 0.05 according to the Tukey test.

**Figure 3 plants-12-00689-f003:**
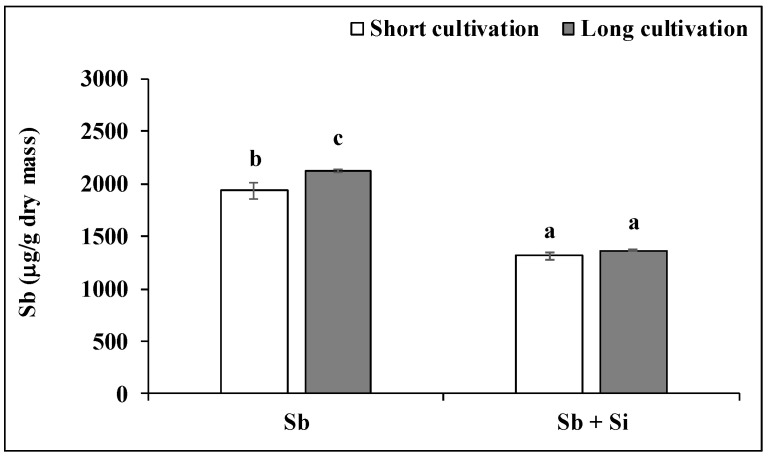
Concentration of Sb (μg g^−1^ dry mass) in poplar calli after the short and long cultivations. The data are presented as the means ± standard error (*n* = 3). Sb treatment—basal medium + Sb (0.2 mM); Sb + Si treatment—basal medium + Sb (0.2 mM) and Si (5 mM). Different letters above bars denote statistically significant differences in the parameters between the treatments at *p* < 0.05 according to the Tukey test.

**Figure 4 plants-12-00689-f004:**
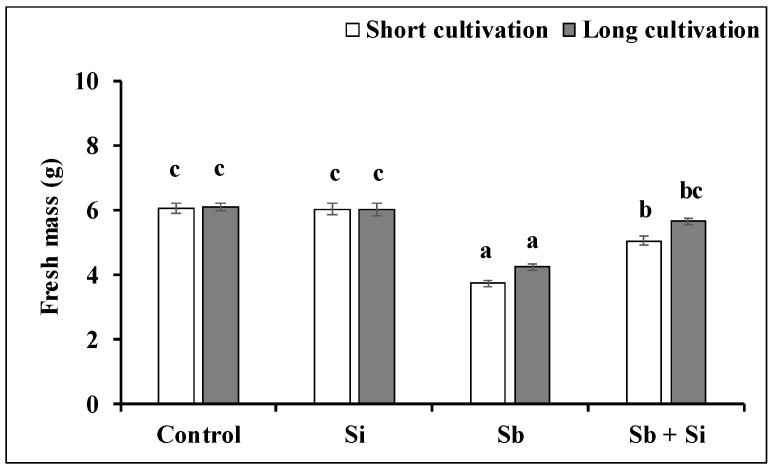
Fresh mass (g) of the poplar calli after the short and long cultivations. The data are presented as the means ± standard error (*n* = 30). Control—basal medium; Si treatment—basal medium + Si (5 mM); Sb treatment—basal medium + Sb (0.2 mM); Sb + Si treatment—basal medium + Sb (0.2 mM) and Si (5 mM). Different letters above bars denote statistically significant differences in the parameters between the treatments at *p* < 0.05 according to the Tukey test.

**Figure 5 plants-12-00689-f005:**
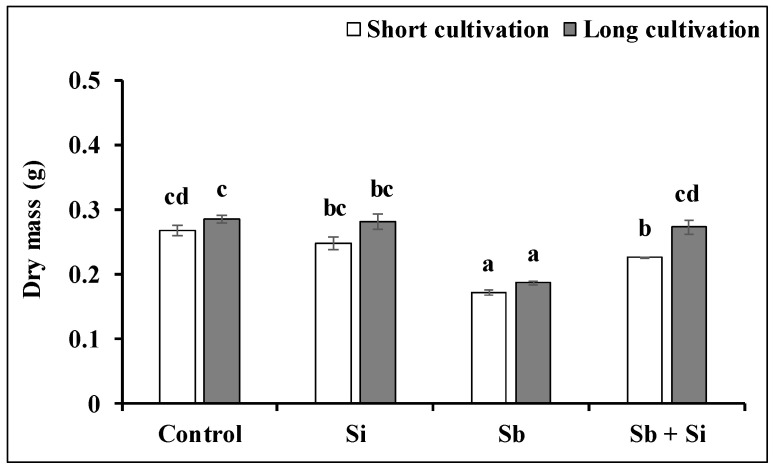
Dry mass (g) of the poplar calli after the short and long cultivations. The data are presented as the means ± standard error (*n* = 30). Control—basal medium; Si treatment—basal medium + Si (5 mM); Sb treatment—basal medium + Sb (0.2 mM); Sb + Si treatment—basal medium + Sb (0.2 mM) and Si (5 mM). Different letters above bars denote statistically significant differences in the parameters between the treatments at *p* < 0.05 according to the Tukey test.

**Figure 6 plants-12-00689-f006:**
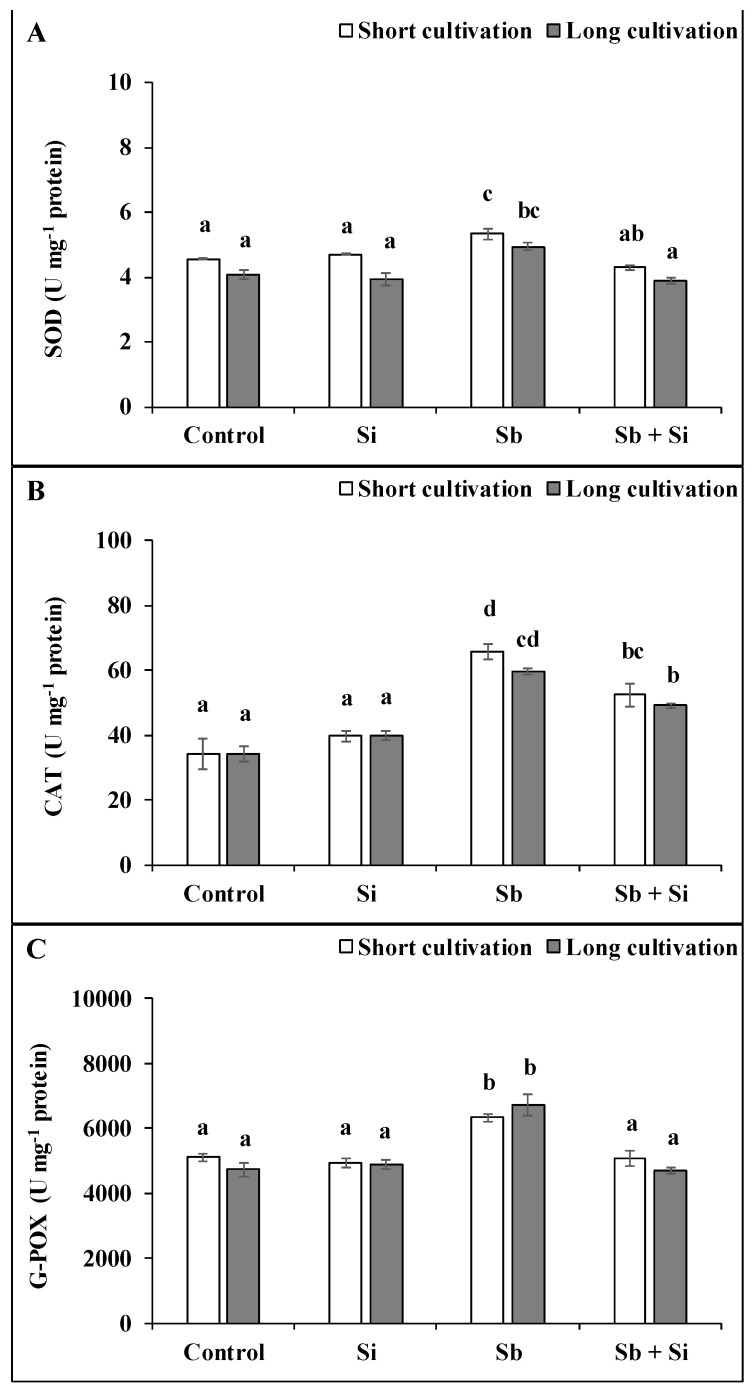
Activity of SOD (superoxide dismutase) (**A**), CAT (catalase) (**B**), G-POX (guaiacol peroxidase) (**C**) in the poplar calli after the short and long cultivations. The data are presented as the means ± standard error (*n* = 6). Control—basal medium; Si treatment—basal medium + Si (5 mM); Sb treatment—basal medium + Sb (0.2 mM); Sb + Si treatment—basal medium + Sb (0.2 mM) and Si (5 mM). Different letters above bars denote statistically significant differences in the parameters between the treatments at *p* < 0.05 according to the Tukey test.

**Table 1 plants-12-00689-t001:** Concentration of chlorophyll *a* (μg g^−1^ fresh mass), chlorophyll *b* (μg g^−1^ fresh mass), and carotenoids (μg g^−1^ fresh mass) in poplar calli after the short and long cultivations. The data are presented as the means ± standard error (*n* = 3). Control—basal medium; Si treatment—basal medium + Si (5 mM); Sb treatment—basal medium + Sb (0.2 mM); Sb + Si treatment—basal medium + Sb (0.2 mM) and Si (5 mM). Different letters (a, b, c, d) denote statistically significant differences in the parameters between the treatments at *p* < 0.05 according to the Tukey test.

		Photosynthetic Pigments (μg g^−1^ Fresh Mass)								
		Chlorophyll *a*			Chlorophyll *b*			Carotenoids		
Control	short cultivation	127.92	±	1.46 ^cd^	48.01	±	1.50 ^bc^	37.75	±	0.64 ^d^
Si		125.39	±	2.23 ^bc^	44.23	±	0.44 ^b^	36.24	±	1.24 ^d^
Sb		42.20	±	4.38 ^a^	15.14	±	15.14 ^a^	13.53	±	1.05 ^a^
Sb + Si		111.21	±	3.83 ^b^	39.69	±	0.73 ^b^	29.99	±	0.46 ^c^
Control	long cultivation	139.87	±	0.95 ^cd^	58.05	±	2.62 ^d^	27.47	±	1.01 ^bc^
Si		142.38	±	1.45 ^d^	61.61	±	1.85 ^d^	30.91	±	0.93 ^c^
Sb		30.14	±	4.73 ^a^	14.78	±	1.69 ^a^	11.24	±	1.09 ^a^
Sb + Si		113.12	±	1.91 ^b^	54.32	±	2.79 ^cd^	24.33	±	0.79 ^b^

**Table 2 plants-12-00689-t002:** Concentration of macronutrients (mg g^−1^ dry mass) in the poplar calli after the short and long cultivations. The data are presented as the means ± standard error (*n* = 3). Control—basal medium; Si treatment—basal medium + Si (5 mM); Sb treatment—basal medium + Sb (0.2 mM); Sb + Si treatment—basal medium + Sb (0.2 mM) and Si (5 mM). Different letters (a, b, c) denote statistically significant differences in the parameters between the treatments at *p* < 0.05 according to the Tukey test.

Concentration of the Macroelements (mg/g Dry Mass)													
		P			K			Ca			Mg		
Control	short cultivation	2.97	±	0.06 ^bc^	54.23	±	0.22 ^c^	6.31	±	0.02 ^b^	2.27	±	0.03 ^bc^
Si		2.93	±	0.05 ^bc^	52.43	±	0.70 ^bc^	6.33	±	0.06 ^b^	2.23	±	0.01 ^b^
Sb		2.49	±	0.02 ^a^	48.47	±	0.41 ^a^	5.79	±	0.04 ^a^	2.05	±	0.03 ^a^
Sb + Si		2.86	±	0.06 ^bc^	52.03	±	0.09 ^bc^	6.19	±	0.04 ^ab^	2.24	±	0.02 ^b^
Control	long cultivation	3.14	±	0.10 ^c^	53.10	±	0.51 ^bc^	6.47	±	0.19 ^bc^	2.39	±	0.06 ^c^
Si		3.06	±	0.07 ^bc^	52.27	±	0.27 ^bc^	6.91	±	0.12 ^c^	2.52	±	0.03 ^d^
Sb		2.46	±	0.04 ^a^	47.13	±	1.10 ^a^	5.83	±	0.07 ^a^	2.08	±	0.01 ^a^
Sb + Si		2.79	±	0.02 ^b^	51.50	±	0.38 ^b^	6.33	±	0.06 ^b^	2.30	±	0.01 ^bc^

**Table 3 plants-12-00689-t003:** Concentration of micronutrients (μg g^−1^ dry mass) in the poplar calli after the short and long cultivations. The data are presented as the means ± standard error (*n* = 3). Control—basal medium; Si treatment—basal medium + Si (5 mM); Sb treatment—basal medium + Sb (0.2 mM); Sb + Si treatment—basal medium + Sb (0.2 mM) and Si (5 mM). Different letters (a, b, c, d, e) denote statistically significant differences in the parameters between the treatments at *p* < 0.05 according to the Tukey test.

Concentration of the Microelements (μg/g Dry Mass)													
		Fe			Mn			Zn			Cu		
Control	short cultivation	155.67	±	3.71 ^bc^	411.33	±	4.26 ^c^	107.80	±	5.96 ^b^	1.3	±	0.01 ^cd^
Si		149.00	±	4.62 ^b^	412.00	±	0.58 ^c^	107.60	±	5.06 ^b^	1.3	±	0.01 ^cd^
Sb		129.33	±	2.60 ^a^	343.00	±	2.89 ^a^	86.00	±	1.25 ^a^	1.1	±	0.02 ^a^
Sb + Si		147.33	±	3.18 ^b^	392.67	±	0.88 ^bc^	104.80	±	1.51 ^b^	1.3	±	0.01 ^bc^
Control	long cultivation	149.33	±	4.67 ^b^	459.33	±	1.76 ^d^	114.57	±	0.67 ^b^	1.5	±	0.02 ^de^
Si		145.33	±	1.20 ^b^	445.00	±	5.03 ^d^	105.40	±	1.62 ^b^	1.5	±	0.09 ^e^
Sb		129.00	±	1.73 ^a^	359.00	±	5.51 ^a^	88.80	±	1.73 ^a^	1.1	±	0.01 ^ab^
Sb + Si		168.33	±	2.03 ^c^	385.33	±	8.01 ^b^	105.20	±	1.10 ^b^	1.4	±	0.01 ^cd^

## Data Availability

The data presented in this study are available on reasonable request from the corresponding author.

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
