# Peer review of "Silicon Actuates Poplar Calli Tolerance after Longer Exposure to Antimony"

_plants, 2023, doi:10.3390/plants12030689_

Round 1

Reviewer 1 Report

The authors of the manuscript titled “Silicon actuates poplar calli tolerance after longer exposure to antimony.” provide an overview on poplar callus (prepared from Populus alba L. var. pyramidallis) was used to determine the involvement of Si in the defence mechanisms against Sb stress. To elucidate the Sb and Si effects, the concentration of Si and Sb (1); the fresh and dry mass (2); the concentration of the photosynthetic pigments (3); the activity of the antioxidant enzymes (4), as well as the concentration of the selected mineral nutrients (5) in the poplar callus were determined. This is a well-written research article and I anticipate that the manuscript should be of great interest. Before recommending this article for publication, there are some shortcomings that should be resolved.

Abstract

The authors are requested to put the quantitative data of the studied parameters in the abstract section of the manuscript.

Introduction

This section of the manuscript is well-written.

Material and methods

Like the introduction, this section is also too much lengthy. Therefore, the authors are requested to avoid unnecessary explanations.

Results

The authors are requested to dissect Figure 2, and Figure 3 which will help the reader to see the lower values of the studied parameters under different treatments.

The authors are also requested to put Table 1 after the end of “The influence of Sb and/or Si on the concentration of photosynthetic pigments”.

Furthermore, make a single Figure of the activity of antioxidant enzymes.

Discussion and Conclusion

These sections are well-written.

Author Response

We would like to thank the reviewer for his comments and for giving us the opportunity to improve the scientific quality of our manuscript.

Reviewer (1):

The authors of the manuscript titled “Silicon actuates poplar calli tolerance after longer exposure to antimony.” provide an overview on poplar callus (prepared from Populus alba L. var. pyramidallis) was used to determine the involvement of Si in the defence mechanisms against Sb stress. To elucidate the Sb and Si effects, the concentration of Si and Sb (1); the fresh and dry mass (2); the concentration of the photosynthetic pigments (3); the activity of the antioxidant enzymes (4), as well as the concentration of the selected mineral nutrients (5) in the poplar callus were determined. This is a well-written research article and I anticipate that the manuscript should be of great interest. Before recommending this article for publication, there are some shortcomings that should be resolved.

Abstract

The authors are requested to put the quantitative data of the studied parameters in the abstract section of the manuscript.

Response: The quantitative data was added to the abstract.

Introduction

This section of the manuscript is well-written.

Response: Thank you for your positive review.

Material and methods

Like the introduction, this section is also too much lengthy. Therefore, the authors are requested to avoid unnecessary explanations.

Response: The Material and methods were shortened.

Results

The authors are requested to dissect Figure 2, and Figure 3 which will help the reader to see the lower values of the studied parameters under different treatments.

Response: Thank you for your suggestion. However, the Sb/Si accumulation in Figure 2 (for the Control and the Sb treatment) and Figure 3 (for the Control and Si treatment), was too low or not detected in the callus tissue. Therefore, we dissected the Figures 2 and 3, and only the higher values are remaining in the graphs. The statistics and figure legends were corrected as well.

The authors are also requested to put Table 1 after the end of “The influence of Sb and/or Si on the concentration of photosynthetic pigments”.

Response: The Table 1 was moved after the section of “The influence of Sb and/or Si on the concentration of photosynthetic pigments”.

Furthermore, make a single Figure of the activity of antioxidant enzymes.

Response: The Figures of antioxidant enzymes were merged into one Figure.

Discussion and Conclusion

These sections are well-written.

Response: Thank you for your positive review.

Reviewer 2 Report

I checked your manuscript and described comments below.

Sb is widely used as an industrial material, and its soil pollution has become a problem.

This paper does a very good job of activating the poplar calli tolerance after prolonged exposure of silicon to Sb.

As a result, we have provided important suggestions regarding the use of Si.

I don't think this paper has any major mistakes or grammatical problems.

Author Response

We would like to thank the reviewer for his comments and for giving us the opportunity to improve the scientific quality of our manuscript.

Reviewer (2):

I checked your manuscript and described comments below.

Sb is widely used as an industrial material, and its soil pollution has become a problem.

This paper does a very good job of activating the poplar calli tolerance after prolonged exposure of silicon to Sb.

As a result, we have provided important suggestions regarding the use of Si.

I don't think this paper has any major mistakes or grammatical problems.

Response: Thank you for your positive review.

Reviewer 3 Report

Dear authors

The manuscript is well written and designed well, minor comments were included:

In the abstract, line 13, SOD, CAT, G-POX should be written without abbreviation when mentioned for the first time. 

Add brief knowledge on the effect of metalloids on calli development.

Improve the aim of the work.

Why do the authors apply long- and short-duration experiments? 

The conclusion should be improved as well. 

Author Response

We would like to thank the reviewer for his comments and for giving us the opportunity to improve the scientific quality of our manuscript.

Reviewer (3)

The manuscript is well written and designed well, minor comments were included:

In the abstract, line 13, SOD, CAT, G-POX should be written without abbreviation when mentioned for the first time. 

Response: The abbreviation in the Abstract was replaced by the full name of enzymes.

Add brief knowledge on the effect of metalloids on calli development.

Response: We added more information to the text.

Improve the aim of the work.

Response: The aims were improved.

Why do the authors apply long- and short-duration experiments? 

Response: The long and short duration experiments were conducted to clarify the impact of Sb and Si on the tissue over time. It is known that the first reactions of the plant tissues are usually more abrupt and the damage caused by the toxic metal is more significant than those that accumulate during longer exposure. This is even more evident for the tolerant plant species, which usually respond to stress by the activation of defence mechanisms. Moreover, Si is proposed as a biostimulant usable for phytoremediation; hence, we aimed to elucidate its effects over longer exposure to Sb.

The conclusion should be improved as well. 

Response: The conclusion was modified and improved as suggested.

Reviewer 4 Report

Comments are enclosed.

Author Response

We would like to thank the reviewer for his comments and for giving us the opportunity to improve the scientific quality of our manuscript.

Reviewer (4)

Lines 36-43 page 1: Statement: In general, Sb in the environment is predominantly present in two forms – antimonate (SbV) or antimonite (SbIII) – while the second one is considered as the chemical form more easily absorbed by plants and more harmful to plants [9,11]. The leakage of harmful amounts of Sb to the environment causes inhibition of plant growth and biomass production, as well as inhibition of photosynthesis [12], and induces leaf chlorosis and necrosis [13]. Antimony might negatively influence the uptake of essential nutrients and disturb the synthesis of some metabolites [14]. It has been also reported that Sb can increase the peroxidation of membrane lipids [1] and cause severe oxidative stress [15, 16].

Remarks: In the above-mentioned paragraph, only possibilities of antimony to be harmful are mentioned. Rather the prior impacted reports or studies which have resulted in damage to human health and environment could have been more beneficial to the topic. Hence, kindly revise this para with recent evidence and reports of antimony hazards to plants and environment.

Response: We added more information to the text.

Lines 51-52 page 2: Statement: Multiple studies confirmed that Si increases the fitness of plants [21, 22, and 23]; however, its exact action in plants is still not known.

Remarks:  Please remove the highlighted part as role of silicon in plants is widely studied and understood.

Response: We removed the highlighted part.

Remarks on overall manuscript • The study of callus from of poplar plant with Si and Sb might give different results in field conditions due to external factors which influence the growth of plants. • The conclusions drawn as per these in vitro studies are based on the results of experiments in controlled conditions, but these deductions are required to be replicated in field conditions to assess the significance of this hypothesis. • If possible, Author can continue research in field conditions also

Response: Thank you for your interesting suggestion. However, here we focused on the effect of Sb and Si on the poplar callus. We chose to work with callus because they are convenient for the study of the changes at the tissue level and calli cells are not lignified. Hence, they are suitable experimental material for the evaluation of the defensive mechanisms of plants grown under stress conditions. In our future studies, we are discussing the possibilities to conduct similar experiments in field conditions to compare them with results from in vitro conditions.

Remarks: Kindly include the following portion in bibliography 1. https://doi.org/10.1007/s11033-021-06469-9

Response: Thank you for the proposed publication but after carefully reading it, we concluded that it is not suitable for our manuscript.

Reviewer 5 Report

The paper „Silicon actuates poplar calli tolerance after longer exposure to  antimony” is current and very well structurate.

In this study the authors highlighted the use of the  poplar callus (prepared from Populus alba L. var. pyramidallis) to determine the involvement of Si in the defence mechanisms against Sb stress.

I propose to publish the paper  after the expanding of  the introduction with more studies in the field.

Author Response

We would like to thank the reviewer for his comments and for giving us the opportunity to improve the scientific quality of our manuscript.

Reviewer (5)

The paper „Silicon actuates poplar calli tolerance after longer exposure to  antimony” is current and very well structurate.

In this study the authors highlighted the use of the  poplar callus (prepared from Populus alba L. var. pyramidallis) to determine the involvement of Si in the defence mechanisms against Sb stress.

I propose to publish the paper after the expanding of the introduction with more studies in the field.

Response: The introduction was expanded as proposed.

Round 2

Reviewer 1 Report

I accept the paper in its present form now. The authors answered my queries well.